# Occurrence and Source Identification of Polychlorinated Dibenzo-p-dioxins and Dibenzofurans and Polychlorinated Biphenyls in Surface Sediments from Liangshui River in Beijing, China

**DOI:** 10.3390/ijerph192416465

**Published:** 2022-12-08

**Authors:** Honghua Li, Pu Wang, Yongming Ju, Wenjuan Li, Ruiqiang Yang, Gang Li, Wenqiang Ren, Jie Li, Qinghua Zhang

**Affiliations:** 1State Key Laboratory of Environmental Chemistry and Ecotoxicology, Research Center for Eco-Environmental Sciences, Chinese Academy of Sciences, Beijing 100085, China; 2Hubei Key Laboratory of Industrial Fume and Dust Pollution Control, School of Environment and Health, Jianghan University, Wuhan 430056, China; 3Laboratory of Pesticide Environmental Assessment and Pollution Control, Nanjing Institute of Environmental Sciences, Ministry of Ecology and Environment (MEE), Nanjing 210042, China; 4The Key Laboratory of Water and Air Pollution Control of Guangdong Province, South China Institute of Environmental Sciences, Ministry of Ecology and Environment (MEE), Guangzhou 510655, China; 5University of Chinese Academy of Sciences, Beijing 100049, China

**Keywords:** PCDD/Fs, PCBs, sediment, Liangshui River

## Abstract

Polychlorinated dibenzo-p-dioxins and dibenzofurans and polychlorinated biphenyls were measured in the surface sediments of Liangshui River, the second largest drainage river in Beijing, China. The sum concentrations of polychlorinated dibenzo-p-dioxins and dibenzofurans and polychlorinated biphenyls ranged from 3.5 to 3019 (mean value: 184) pg g^−1^ dry weight and from 319 to 5949 (mean value: 1958) pg g^−1^ dry weight, and the corresponding World Health Organization toxic equivalent quantity values were 0.0011–5.1 pg TEQ g^−1^ dry weight and 0.0074–1.4 pg TEQ g^−1^ dry weight, respectively. The spatial distributions of polychlorinated dibenzo-p-dioxins and dibenzofurans and polychlorinated biphenyls showed increasing trends from urban area and development area to suburb. Principal component analysis revealed that polychlorinated dibenzo-p-dioxins and dibenzofurans contamination in the sediments may originate from pentachlorophenol and sodium pentachlorophenate and municipal solid waste incineration. Regarding polychlorinated biphenyls, the steel industry, combustion processes and usage of some commercial polychlorinated biphenyl products were identified as the major sources. The emission from a former steel plant could be the main contributor to polychlorinated biphenyls in urban areas. The mean value of the total toxic equivalent quantities in the sediment samples exceeded the Canadian interim sediment quality guidelines. Long-term wastewater irrigation increases the load of sediment-bound pollutants in agricultural soil and may pose potential ecological risks to crops and human health.

## 1. Introduction

Polychlorinated dibenzo-p-dioxins and dibenzofurans (PCDD/Fs) and polychlorinated biphenyls (PCBs) are well-known persistent organic pollutants (POPs). These compounds have caused global concerns in the past decades due to their high toxicity, difficult degradation and bioaccumulative properties and have been listed into the Stockholm Convention on POPs since 2001. PCDD/Fs are formed unintentionally in combustion processes, such as waste incineration, forest fires, metal production, power generation and burning of fossil fuel [1,2]. PCDD/Fs are also released to the environment as unwanted byproducts from various industrial processes, such as metal smelting, pulp bleaching and production of chlorinated chemicals including chloro-phenols and chloro-organic pesticides [3,4]. PCBs were mainly produced as industrial chemicals in the last century, such as coolants and insulators in transformers and capacitors, plasticizers in printers, paints and rubber sealant [5]. PCBs can enter the environment during their manufacturing and usage, and continue to be released in electronic waste recycling even though the uses of PCBs were banned worldwide in 1980s. Additionally, the byproducts of combustion are also the important unintentional sources of PCBs in the environment [6]. The major pathways of PCDD/Fs and PCBs to aquatic environment include atmospheric transport and deposition, industrial and domestic discharge, storm and river inputs [7,8]. Due to their hydrophobic properties [9], these compounds are strongly adsorbed on particulate matters in water body, and mostly deposited into sediments. Aquatic organisms are exposed to these compounds through direct contact and ingestion, which may pose a potential threat to aquatic organisms and even human health through aquatic food chains.

Liangshui River (LSR) is the second largest drainage river in Beijing. It rises in the outlet of the sewage treatment plant (STP) of the steel plant of Beijing, and flows southeast into the North Canal main stream through Fengtai, Daxing, and Tongzhou Districts. It stretches 68 km with a basin extending over 600 square kilometers. There are more than one thousand drain outlets along the river banks, 70% of which work all the year round. Annually, LSR receives about three hundred million tons of treated or untreated domestic sewage and industrial effluent, accounting for 20% of the total discharge in Beijing’s central region. The regulation projects of LSR have been carried out by the Beijing government since the early 1990s, including controlling pollution sources, building or expanding wastewater treatment plants (WWTP) and promoting sewage pipe network management, and the water quality was therefore improved in the early years. However, due to the rapid population growth, the decreasing precipitation, the lag in the WWTP expansion and the insufficient sewage interception and treatment capacities, LSR was subjected to severe pollution.

Owing to a water shortage in the megacity of Beijing, wastewater is still one of the important non-conventional sources for many rivers such as LSR, and therefore is used widely for agricultural purposes by farmers along many rivers. The LSR basin, the main agricultural area in Beijing, is such a wastewater irrigation area. Pollutants in the aquatic environment would be inevitably transferred to land [10,11,12] through irrigation, and subsequently to crops and humans. Several investigations have reported that LSR was influenced by the discharge of drainage, with polycyclic aromatic hydrocarbons (PAHs) [13], pesticides [14] and antibiotics [15] being observed. These studies focused on the contamination of the surface water in LSR. Information concerning PCDD/Fs and PCBs in the sediment of LSR is deficient, and the pollution sources and potential risks of wastewater irrigation remain unknown. Hereby, we present a comprehensive investigation on PCDD/Fs and PCBs in sediments from the Beijing drainage river to determine the occurrence and spatial distribution characteristics of these compounds, identify the possible sources of PCDD/Fs and PCBs according to their composition of congeners and homologues, and assess the sediment quality of LSR. This study aims to acquire basic records of PCDD/Fs and PCB contamination in sediments from the metropolitan drainage river, gain insight into the risk assessment of wastewater irrigation and provide assistance to the administration in wastewater irrigation, wastewater treatment and management of municipal drainage rivers.

## 2. Materials and Methods

### 2.1. Study Sites and Sampling

The LSR basin is situated in the southeast of Beijing. It is characterized by a typical temperate and continental monsoon climate with an average annual temperature of 11.3 °C and annual precipitation of 508.4 mm, and the rainfall is concentrated in July–September. The sampling campaign was carried out in the flood season of 2013, and the depth of LSR was up to 3 m. Samples were collected from 27 sites, among which 22 sites were along the LSR main stream, and the other five sites were on the tributaries, including the Xingfeng River, the Fenggangjian River and the Xiaotaihou River. The sampling map is shown in Figure 1. L01–L05 are in Beijing’s urban area; L06 and L22–L25 are located in the Economic Technological Development Area; and L07–L21 and L27 are located in the suburbs. In total, 27 surface sediments (0–5 cm) were sampled with a stainless steel grab sampler. All the samples were sealed in clean polyethylene bags, carried to the laboratory immediately and then stored at –18 °C. Freeze-dried samples were ground into a fine powder, and then sieved through a 16-mesh sieve. After being homogenized, the samples were kept under –18 °C prior to further pretreatment.

### 2.2. Chemicals

^13^C_12_-labeled standard solutions of PCDD/Fs (purity > 99.9%) (calibration standard EPA 1613-CVS, surrogate standard EPA 1613-LCS and injection standard EPA 1613-IS) and PCBs (purity > 99.9%) (calibration standard EPA 68C-CVS, surrogate standard EPA 68C-LCS and injection standard EPA 68C-IS) were all provided by Wellington Laboratories (Guelph, Canada). Pesticide grade dichloromethane (DCM), n-hexane and toluene were purchased from Fisher (Fair Lawn, NJ, USA). Analytical grade anhydrous sodium sulfate (Na_2_SO_4_), sodium hydroxide (NaOH) and concentrated sulfuric acid (H_2_SO_4_) were supplied by Beijing Chemical Factory (Beijing, China). Silica gel 60 (0.063–0.100 mm), basic alumina, active carbon and diatomaceous earth were obtained from Merck (Darmstadt, Germany), Sigma-Aldrich (St. Louis, MI, USA), Supelco (Bellefonte, PA, USA) and DIONEX (Sunnyvale, CA, USA) respectively. The preparation methods of 30% (*w*/*w*) sulfuric acid-impregnated silica gel and 1.2% (*w*/*w*) sodium hydroxide impregnated silica gel have been reported previously [16].

### 2.3. Sample Preparation

Approximately 5 g of freeze-dried sediment sample was spiked with ^13^C_12_-labeled PCDD/Fs (EPA 1613-LCS) and ^13^C_12_-labeled PCBs (EPA 68C-LCS) and homogenized with 10 g of Na_2_SO_4_. The sample was extracted with DCM/n-hexane (1:1 *v*/*v*) using an accelerated solvent extractor (ASE 300, Dionex, Sunnyvale, CA, USA). The extraction was carried out at 150 °C and a pressure of 1500 psi with 7 min of heating time followed by 8 min of static time. The flush volume was 60%, the purge time was 120 s, and two static cycles were performed. After extraction, granulated activated copper was added in the receiving flasks to remove elemental sulfur. The extracted solution was concentrated to 1 mL with a rotary evaporator (Heidolph, Schwabach, Germany) and transferred into a multilayer silica gel column (10 mm i.d.) for cleanup. The silica gel column was filled with 1 g of neutral silica gel, 4 g of 1.2% NaOH impregnated silica gel, 1 g of neutral silica gel, 8 g of 30% H_2_SO_4_-impregnated silica gel and anhydrous Na_2_SO_4_ (2 cm height) (from bottom up). The column was conditioned with 80 mL of n-hexane and then eluted with 100 mL of n-hexane after the extract was loaded. The eluate was reduced to about 1 mL and applied to basic alumina column for further purification. The column was packed with 6 g of basic alumina at the bottom and 1 cm of Na_2_SO_4_ on the top and conditioned with 50 mL n-hexane. The target compounds were eluted using 40 mL of DCM/n-hexane (1:1 *v*/*v*). After being concentrated to 1 mL, the eluate was passed through a carbon column (1.5 g) filled with 18% activated carbon impregnated diatomaceous earth to separate PCDD/Fs and PCBs. The carbon column was conditioned with 10 mL of toluene and 10 mL of n-hexane, respectively, and then eluted with 50 mL of n-hexane (the first fraction contains PCBs) and followed by 80 mL of toluene (the second fraction contains PCDD/Fs). The two final fractions were evaporated and solvent exchanged to 20 μL of nonane in sample vials. Then, 1 ng of the injection standards (1613-IS and 68C-IS) was added before instrumental analysis.

### 2.4. Instrumental Analysis

PCDD/Fs and PCBs were determined using a high-resolution gas chromatographer (Agilent 6890N, Wilmington, NC, USA) coupled with high-resolution mass spectrometer (AutoSpec Ultima, Waters Micromass, Wilmslow, UK) (HRGC/HRMS). Chromatographic separation was achieved on a DB-5MS fused silica capillary column (60 m × 0.25 mm i.d. × 0.25 μm, Agilent, USA). The mass spectrometer was operated in positive electron impact (EI^+^) and the targets were analyzed in voltage selective ion records (VSIR) mode at a mass resolution of above 10,000. The electron energy was 35 eV, and the ion source and transfer line were kept at 270 °C. Helium was used as the carrier gas at a flow rate of 1 mL min^−1^. The injection port temperature was held at 270 °C and 1 μL of the sample was injected into the GC in splitless mode. The GC oven temperature was programmed as follows: for PCBs, the initial temperature was 120 °C, held for 1 min, and then increased to 150 °C at a rate of 30 °C min^−1^, ramped to 300 °C at 2.5 °C min^−1^ and held for 1 min; for PCDD/Fs, 150 °C for 3 min, 20 °C min^−1^ to 230 °C and held for 18 min, increased to 235 °C at 5 °C min^−1^ and held for 10 min, and finally 4 °C min^−1^ to 330 °C and held for 3 min. Seventeen 2,3,7,8-substituted PCDD/F congeners, twelve dioxin-like PCB congeners, six indicator PCB congeners and six other PCB congeners were determined.

### 2.5. Quality Assurance and Quality Control (QA/QC)

Determination of PCDD/Fs and PCBs was based on the US EPA method 1613B and US EPA method 1668C. Quantification of target analytes was conducted by the isotope dilution method using relative response factors (RRFs) determined from calibration curves. EPA 1613-LCS and EPA 68C-LCS were used for the quantitation of PCDD/Fs and PCBs, respectively, and the recoveries of these surrogate standards were calculated by EPA 1613-IS and EPA 68C-IS. A laboratory blank was tested for every batch of 10 samples. The limit of detection (LOD) was defined as three times the signal-to-noise ratio. The recoveries of ^13^C_12_-labeled surrogate standards were 63–85% for PCDD/Fs and 53–138% for PCBs, and the isotopic ratios of the two main ion pairs were within the limits, which met the requirements of the EPA methods. The LODs ranged between 0.04–9.38 pg g^−1^ and 0.04–7.4 pg g^−1^ for PCDD/Fs and PCBs, respectively. The PCDD/F concentrations in the laboratory blanks were below the LODs. Several PCB congeners (i.e., PCB-28 and PCB-52) were consistently detected at a relatively low level in the blanks, and the reported concentrations were therefore blank corrected.

## 3. Results and Discussion

### 3.1. Concentrations of PCDD/Fs and PCBs

The concentrations of PCDD/Fs and PCBs are summarized in Table 1 and Table 2, respectively. PCDD/Fs and PCBs were detected in all the samples. The sum PCDD/F concentrations (Σ_17_PCDD/Fs) ranged from 3.5 to 3019 pg g^−1^ dry weight (dw), with an average value of 184 ± 559 pg g^−1^ dw (mean ± standard deviation) and a median value of 61.5 pg g^−1^ dw. The sum concentrations of PCBs (Σ_24_PCBs) varied within the range of 319–5949 pg g^−1^ dw, with a mean of 1958 ± 1443 pg g^−1^ dw and a medium of 1297 pg g^−1^ dw. The corresponding World Health Organization toxic equivalent quantity (WHO-TEQ) values were in the range of 0.0011–5.1 pg TEQ g^−1^ dw and 0.0074–1.4 pg TEQ g^−1^ dw for PCDD/Fs and PCBs, respectively. In order to be comparable with previous studies (Appendix A), WHO-TEQ calculated based on WHO_2005_-TEFs and WHO_1998_-TEFs [9] of PCDD/Fs and PCBs are also given in Table 1 and Table 2.

The PCDD/F levels in the sediments of LSR were 2–3 orders of magnitude lower than those in the drainage river in North China [17], the industrialized River Elbe in Germany [18] and the Norwegian Grenlandsfjords [19], while they were the same order of magnitude of those in the Liaohe River [20], the Pearl River and the Yangtze River in China [21], the Great Lakes of North America [22], the Black Sea [23] and Lake Victoria in East Africa [24], and higher than those in the Yellow River in China [21], Lake Shihwa in Korea [25] and the southern North Sea and the English Channel [26].

For PCBs, the results in the present study were 1–2 orders of magnitude lower than those in the Dagu Drainage River [17], the Liaohe River [20], the Pearl River Delta in China [27] and the Houston Ship Channel in the United States [28]. Our results were comparable with the studies on the Haihe River basin in China. Liu et al. [17] reported the PCB levels ranging between 1142 and 7473 (mean value: 3077) pg g^−1^ dw, corresponding to 0.1 to 0.5 (mean value: 0.27) pg TEQ g^−1^ dw. Wang et al. [21] found that PCBs in the surface sediments from the Haihe River varied between 2820 to 5630 (mean value: 3840) pg g^−1^ dw, corresponding to 0.03 to 0.1 (mean value: 0.053) pg TEQ g^−1^ dw. Similar results were also observed in the sediments from other locations in China, such as the Yellow River and the Yangtze River [21], and from other countries and regions including the River Elbe in Germany [18], the pounds in France [29] and the Guaratuba Bay in Brazil [30]. By contrast, our results were one order of magnitude higher than those from the estuaries and coasts [26,31].

### 3.2. Spatial Distribution of PCDD/Fs

The PCDD/F concentrations varied widely among sampling locations (Appendix A). An extremely high level (3019 pg g^−1^ dw) was found at the L20 site, which was 10–900 times higher than those of the other sites. L20, lying at the end of the LSR downstream, was close to the confluence of LSR and the North Canal which is Beijing’s major wastewater channel to the Haihe River. In addition, upstream of the North Canal receives wastewater discharged from the other three main drainage rivers including the Qinghe River, the Bahe River and the Tonghuihe River. Nearly 90% of the treated wastewater and other untreated wastewater in Beijing was discharged into the North Canal [32], which may cause severe contamination of PCDD/Fs. The sediments from L06, L17 and L23 had relatively low levels of PCDD/Fs (<10 pg g^−1^ dw). As shown in Figure 1, these sampling sites were behind the rubber dams built on LSR and its tributaries. L22 and L23 were separated by a rubber dam, and the former had a higher level than the latter. The PCDD/F concentrations in L06 and L17 showed the similar tendencies. A previous study [17] also reported the fluctuation of PCDD/F concentration in the sediments collected from the inside and outside of the floodgate built on the river.

Figure 2 depicted the spatial distribution of PCDD/Fs in the three functional urban areas. L01–L05 were located upstream of LSR in urban area, where there is a large WWTP in Beijing with a treatment capacity of 600,000 tons per day. The mean PCDD/F concentration was 50.6 pg g^−1^ dw in this area and the highest value was found at L02 (102 pg g^−1^ dw). L06 and L22–L25 were in the Yizhuang Economic Technological Development Area, where more than four thousand manufacture enterprises are sited. Nevertheless, the average concentration of PCDD/Fs in the sediments was the lowest (26.1 pg g^−1^ dw), and approximately half of that in the urban area. In recent years, a completed ecological industry system has been established in the Yizhuang Park. The high reusing rate of industry water and reclaimed water improved the utilization efficiency of water resource, and therefore decreased the industrial pollution to the aquatic environment. L26, L27 and L07–L21 in the lower reach of LSR in the suburbs were found to be heavily polluted with a mean PCDD/F level of 270 pg g^−1^ dw. This stretch of LSR is in the suburbs of Beijing, and outside of the sixth Ring Road, where the construction of sewage pipe networks is relatively lagging. Moreover, this area is the site of the Tongzhou industrial cluster featuring fine chemical industry, pharmaceutical industry, material industry and electronic industry. There is only one WWTP with a treatment capacity of 4000 tons per day, and the capacity of wastewater treatment is insufficient. LSR downstream flows through this zone, and the industrial effluent discharge might contribute to PCDD/Fs contamination.

### 3.3. Spatial Distribution of PCBs

L27 was the most heavily polluted site with the PCB concentration of 5948 pg g^−1^ dw. This site is in one of the tributaries of LSR, the Xiaotaihou River. This area is the chemical industry park of Beijing, where some chemical factories, dye plants, coking plants and pharmaceutical factories are grouped. In addition, there is a large landfill of municipal solid waste near the Xiaotaihou River. The heavy pollution of PCBs might come from the discharge of industrial wastewater, emission of exhaust gas and leachate of landfill. The lowest concentration of PCBs (319 pg g^−1^ dw) was detected at the site L06 situated behind a rubber dam. The PCB level decreased sharply from L16 (2838 pg g^−1^ dw) to L17 (1143 pg g^−1^ dw), which might be due to the protection of the dam between the two sites. However, the decreasing trend of PCB concentration was not found in the less polluted sites of L22 and L23 around a rubber dam.

The concentration distribution of PCBs in the three functional urban areas is given in Figure 2. In general, the spatial distribution was similar to that of PCDD/Fs. The PCB levels in the urban area and development area were lower than those in the suburbs. In the urban area, the average concentration was 1023 pg g^−1^ dw and the highest level was detected in L02 (2492 pg g^−1^ dw). At the sites of L06 and L22–L25 in the Yizhuang Economic Technological Development Area, the mean level was 1177 pg g^−1^ dw, which was slightly higher than that of the urban sites. The PCB level in the suburbs was 2463 pg g^−1^ dw on average and showed the same increasing trend as PCDD/Fs. This trend revealed a great contribution of heavy industrial activities in the Beijing chemical industry park and the Tongzhou industrial cluster.

### 3.4. Compositions of PCDD/Fs and PCBs

The compositions of PCDD/Fs and PCBs are summarized in Table 1 and Table 2. The PCDDs/PCDFs ratios in the 27 sediment samples were in the range of 0.55–17, among which two thirds were >1. Generally, the high ratio of PCDDs/PCDFs comes from chlorinated chemical industries, while the low ratio may relate to the high temperature industrial processes and combustion processes [33,34]. As shown in Figure 3, the proportions of PCDD/F congeners were generally similar among the sediment samples. They were characterized by high contribution of OCDD and OCDF, accounting for 77% of the sum PCDD/Fs, and followed by 1,2,3,4,6,7,8-HpCDF and HpCDD, together accounting for more than 14% of the sum PCDD/Fs. The proportions of tetra- through hexa- CDD/F congeners were generally less than 10% or not detected at the light polluted sites, such as L06, L17 and L23 behind the rubber dams. The relatively higher contribution of high-chlorinated PCDD/F congeners has also been found in other studies. Nieuwoudt et al. [35] reported that the homologue contributions of PCDD/Fs in soils and sediments of central South Africa consisted of OCDD > OCDF > HpCDD > HpCDF. The survey on PCDD/Fs in the surface sediments from Korea [36] found that the OCDD contribution was > 50%, followed by OCDF, 1,2,3,4,6,7,8-HpCDF and 1,2,3,4,6,7,8-HpCDD. OCDD and 1,2,3,4,6,7,8-HpCDD accounted for more than 78% of the sum concentration of PCDD/Fs in sediment and biota of Portugal [37]. The congener profiles of PCDD/Fs in sediments from the lakes and the coastal sea areas in Shandong Peninsula of China [38] were also characterized by high OCDD, OCDF and 1,2,3,4,6,7,8-HpCDD.

As shown in Table 2, the compositions of the 12 dioxin-like PCBs (dl-PCBs), 6 indicator PCBs and 6 other PCB congeners were quite similar in the samples, accounting for 10%, 40% and 50% of the sum PCB concentrations on average, respectively. The heavier PCBs made greater contributions to the sum concentrations following the order: deca-CB (36%) > Hepta-CBs (20%) > Tri-CBs (15%) > Penta-CBs (11%). PCB-209 and PCB-28 contributed more than 50% of the sum PCBs. Currently, few studies have focused on PCB-209 pollution in the environment due to its less toxic characteristic and not being the common congener in technical PCBs. Guo et al. [39] observed high concentrations of PCB-209 in sewage sludge samples of urban wastewater treatment plants and inferred that the industrial effluent was the major source. Howell et al. [28] reported that the high fraction of PCB-209 in the Houston Ship Channel (HSC) originated from the unusual Aroclor mixtures used in the history of the HSC or contemporary sources from local industries. Wang et al. [12] found that wastewater irrigated cropland contained a high proportion of PCB-209. Though it is difficult to trace the origin of PCB-209 in this study, the relative higher abundance of PCB-209 in industrial park than that in urban area indicates that industrial effluent could be the important source of PCB-209 [6]. Concerning indicator PCBs, PCB-28 was the dominant congener, accounting for 37%. The results were consistent with those observed in the Dianchi Lake [40], seven major river basins in China [21] and WWTP [39]. Figure 3 shows that the contributions of PCB-28 in the sediments from urban area (mean value: 34%) were obviously higher than those from development area and suburb (mean value: 12%), suggesting that municipal sewage is an important source of indicator PCBs. For dl-PCBs, all the samples had the similar congener profiles. PCB-118 was the predominant congener, accounting for 40% of the sum dl-PCBs on average, and followed by PCB-105 (15%) and PCB-77 (13%). These results were similar to those previously reported in sewage sludge [39] and sediments [21,25,35].

### 3.5. Principal Component Analysis Analysis and Source Implication

Emissions from various POPs sources usually show the characteristic profiles of homologues and congeners, and thus the profiles can be used to identify the potential sources of pollutants. In the present study, principal component analysis (PCA) was employed to identify the potential sources of PCDD/Fs and PCBs in LSR. The typical sources of PCDD/Fs (Appendix A) included chlorinated chemical industries, metallurgy industries, combustion processes, wastewater and solid waste treatment [20]. Concentrations of the 17 PCDD/F congeners were normalized to the sum concentrations before data processing. As shown in Figure 4, the first principal components (PC1) and second principal components (PC2) explained 49% and 20% of the total variance, respectively. All the samples were grouped around PCP, PCP-Na and MSWI, suggesting that PCP, PCP-Na and municipal solid waste incineration might be related to the PCDD/Fs pollution in LSR. PCP and PCP-Na used as general pesticides, paddy herbicides and wood and textile preservatives have been produced in China since the late 1950s. As the impurities in technical PCP and PCP-Na, high levels of PCDD/Fs were detected in the areas where PCP and PCP-Na were largely produced and used [17,41,42]. The eastern and southern suburbs of Beijing are the major grain producing areas and the industrial bases of chemical, textile and pharmaceutical industries. It can be inferred that the use of PCP and PCP-Na as pesticides [43], municipal solid waste incineration and the industrial emissions could contribute to the contamination of PCDD/Fs in the LSR basin.

PCBs are a kind of industrial products and are mainly used in power capacitors, transformers and paint additives in China. Approximately 10,000 tons of commercial PCB products were produced in China from 1965 to 1974, in which 9000 tons was trichlorobiphenyl (similar to Arochlor 1242), and 1000 tons was pentachlorobiphenyl (similar to Arochlor 1254) [20]. Electronic waste (E-waste) dismantling and recycling have been reported to be one of the important sources of PCBs in China [16,50]. In addition, industrial activities and combustion processes such as iron ore sintering, steel manufacturing, metallurgy and coal burning also contribute to PCB contamination. Considering the high toxicity of dl-PCBs, PCA was also conducted for the sediment samples in this study and the selected sources (Appendix A) to identify the potential sources. Concentrations of the individual congeners were normalized to the sum dl-PCBs before PCA was performed. As shown in Figure 5, the extracted top three principal components explained 70% of the total variability (28% for PC1, 26% for PC2 and 16% for PC3), and thus can be used to reveal the relationships between the raw data in this study and the suspected sources. Most of the sediment samples had relatively low factor scores (between –1 and 1) in the first three principal components, and were grouped around the steel industry, combustion sources, WWTP, municipal solid waste and some commercial PCB products (Figure 5). However, the sites L06, L22 and L27 had high factor scores in PC2, and thus were far away from other sites. Gaseous emissions from secondary aluminum and copper metallurgies had remarkably higher positive factor scores in PC1 and negative scores in PC2 (Figure 5a), and were obviously separated into another group, indicating that they could not be the major sources of PCBs in LSR. It should be noted that L01–L05 (Figure 5b) were clustered together with ambient air around the steel plant and gaseous emission from the iron ore sintering plant. These samples were all collected from the upstream of LSR where a former large steel plant was located, and the river just originates from the STP of the plant. The PCA results confirmed that the PCB contamination in urban sampling stations may originate from steel industry. It is known that the plant has stopped steelmaking since 2011 and moved away from Beijing. However, our observations suggested that its impact on the local environment still exists. Based on these results, it can be inferred that the PCB contamination in LSR could be mainly attributed to the inputs of the steel industry and combustion processes including coal and hardwood burning [48], and the usage of commercial PCB products also made a contribution to some extent.

### 3.6. Potential Ecotoxicological Risks of PCDD/Fs and PCBs

Sediments provide the natural habitat for aquatic organisms. Acting as the important sinks and repositories of POP cycling, sediments subsequently become a main source of these pollutants in the environment. Due to their high toxicity and great chemical stability, PCDD/Fs and PCBs in sediments may pose long-term exposer risks to aquatic ecosystems and human health. In this study, the toxicity of sediments was estimated by the TEQs calculated using toxic equivalent factors to 2,3,7,8-TCDD. Sediment quality guidelines (SQGs) derived by the Canadian Council of Ministers of the Environment (CCME) [54] were used to identify and evaluate the ecotoxicological risks of PCDD/Fs and PCBs in the sediments from LSR. The guidelines define two ranges of concentrations expressed on a TEQ basis and adjusted by a safety factor of 10: interim sediment quality guidelines (ISQGs) of 0.85 pg TEQ g^−1^ dw, below which the incidence of adverse biological effects rarely occur, and the probable effect level (PEL) of 21.5 pg TEQ g^−1^ dw, above which adverse effects are expected to occur frequently. The TEQ levels presented in the guidelines include concentrations of 2,3,7,8-substituted PCDD/Fs and coplanar PCBs. Figure 6 showed the ISQGs and the total TEQ (the sum TEQ levels of PCDD/Fs and PCBs) in our study. The spatial distribution of the TEQ levels showed site-specific characteristics. Relatively higher TEQ levels were found in the downstream sediments than those in the upstream sediments. The top five values were all from the downstream sediments. As mentioned above, the downstream area of LSR was subject to multidisciplinary industrial effluent discharge and low capacity of wastewater treatment. The results of TEQ levels and spatial distribution (Figure 6 and Figure 2) indicate that industrial inputs were the main sources of PCDD/Fs and PCBs.

None of the total TEQs at the 27 sites exceeded the PEL, while their mean value was above the ISQGs. These results indicate that adverse biological effects would occur occasionally, and the sediments of LSR are considered to be hazards to exposed organisms. In light of the above, it is suggested that continued monitoring and assessment of sediment quality of LSR may be necessary. In addition to the aquatic environment, it should be noted that LSR is the important agricultural water source in Beijing, and has been available for half a century. PCDD/Fs and PCBs in the sediments could be the secondary sources to the overlying water column [12,22], could be transferred to soils and crops by way of irrigation and eventually enter human body through food chains. The data set in the present research should be used as a baseline and reference sites for source identification and risk assessment and management. Further investigations for the ecotoxicological risks of sediment-bound pollutants and effective management strategies of wastewater discharge are urgent.

## 4. Conclusions

In the present study, PCDD/Fs and PCBs were analyzed in the surface sediments collected from LSR. Concentrations of these pollutants were generally comparable to the results reported in other regions globally. Spatial distribution confirmed the impact of the wastewater discharged from industrial activities on occurrence of PCDD/Fs and PCBs. The PCDD/Fs contamination might be attributed to PCP and PCP-Na used as pesticides, as well as the solid waste treatment. While the PCB contamination mainly originated from the mixed sources of the steel industry, combustion processes and, to a lesser extent, some commercial PCB products. The average of the total TEQs at the 27 sites exceeded the ISQGs, and higher TEQ values were observed at the sampling sites from the downstream area.

The data obtained from the present study suggest that the contaminated sediments could be the secondary sources of pollutant emissions to the river. In the water-scare regions, long-term wastewater irrigation increases the load of sediment-bound pollutants on farmland. Moreover, the pollutants accumulated in agricultural soils could be transferred to crops, and in the end, to humans, which should pose potential risks to food safety and public health. The rapid development and population growth increase the demands for water resources, and wastewater is commonly used for agriculture in the regions of water shortages. Therefore, more water quality monitoring, source identification and environmental fate investigations should be carried out to assess the potential risks of pollutants to the environment and human health. Management concerns should be focused on controlling wastewater discharge from industrial sources, increasing the capacities and efficiencies of wastewater treatment, and environmental remediations may also be necessary in the seriously contaminated regions.

## Figures and Tables

**Figure 1 ijerph-19-16465-f001:**
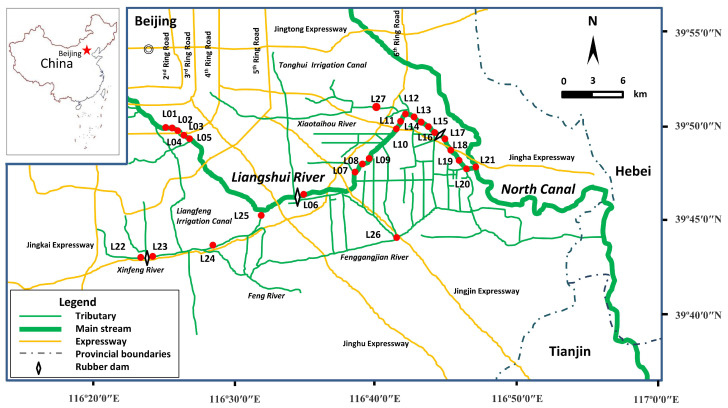
Sampling sites for the surface sediments from Liangshui River (LSR) in Beijing, China.

**Figure 2 ijerph-19-16465-f002:**
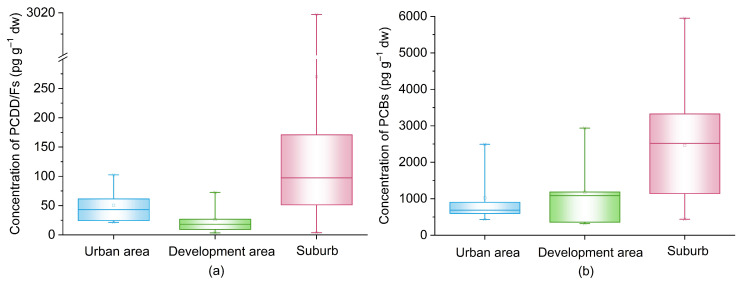
Spatial distribution of polychlorinated dibenzo-p-dioxins and dibenzofurans (PCDD/Fs) (**a**) and polychlorinated biphenyls (PCBs) (**b**) in the three functional urban areas in Beijing. Urban area included 5 sites, L01–L05; development area included 5 sites, L06 and L22–L25; suburb included 17 sites, L07–L21 and L26–L27.

**Figure 3 ijerph-19-16465-f003:**
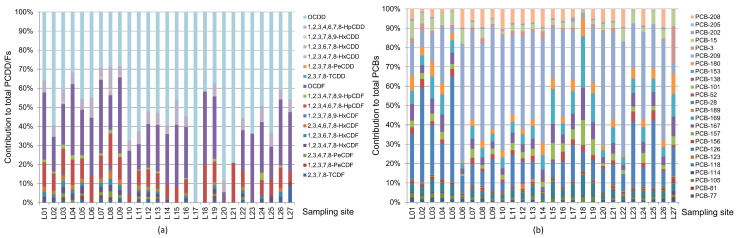
Percentage composition of PCDD/F (**a**) and PCB (**b**) congeners in the sediments from LSR.

**Figure 4 ijerph-19-16465-f004:**
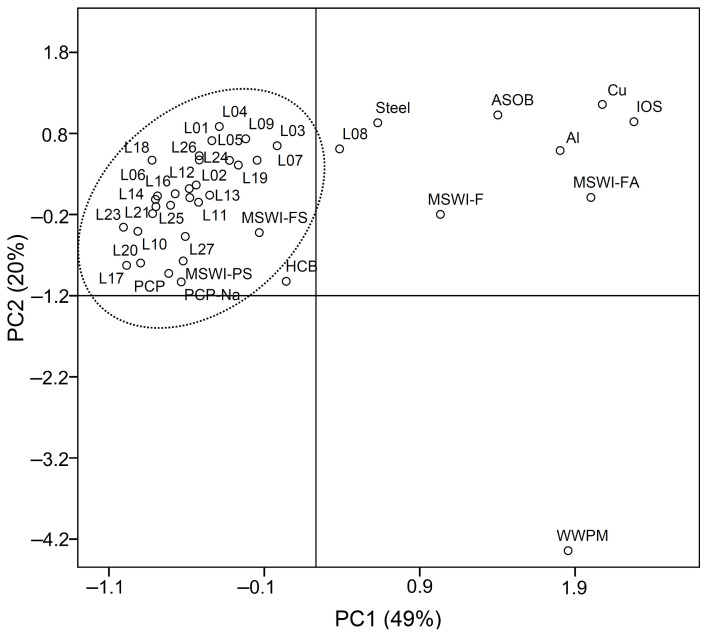
Principal component analysis (PCA) score plot of the 2,3,7,8-substituted PCDD/F congeners in the sediments of LSR and the samples from the typical sources. Abbreviations: PCP, pentachlorophenol [44]; PCP-Na, sodium pentachlorophenate [44]; STEEL, ambient air around a steel plant [45]; Al, gaseous emission from secondary Al metallurgy [20]; Cu, gaseous emission from secondary Cu metallurgy [20]; IOS, gaseous emission from iron ore sintering plants [20]; MSWI-FS, fluvo-aquic soil near to a municipal solid waste incineration (MSWI) plant [46]; MSWI-PS, paddy soil near to a MSWI plant [46]; MSWI-F, flue gas from an MSWI plant [46]; MSWI-FA, fly ash from a MSWI plant [46]; WWPM, waste water from a pulp mill [47]; HCB, housecoal burning [48]; ASOB, agricultural straw open burning [49].

**Figure 5 ijerph-19-16465-f005:**
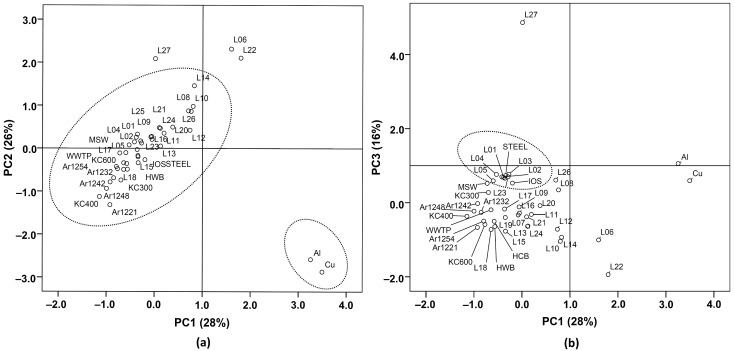
PCA score plot of the dioxin-like PCB (dl-PCB) congeners in the sediments of LSR and the samples from the typical sources; (**a**) PC1 and PC2 score plot; (**b**), PC1 and PC3 score plot. Abbreviations: HCB, housecoal burning [48]; HWB, hardwood burning [48]; KC, brand name of commercial PCB formulations, Kanechlor [51]; Ar, brand name of commercial PCB formulations, Aroclor [52]; Al, gaseous emission from secondary Al metallurgy [20]; Cu, gaseous emission from secondary Cu metallurgy [20]; IOS, gaseous emission from iron ore sintering plants [20]; WWTP, sewage sludge of urban wastewater treatment plants [39]; STEEL, ambient air around a steel plant [45]; MSW, municipal solid waste [53].

**Figure 6 ijerph-19-16465-f006:**
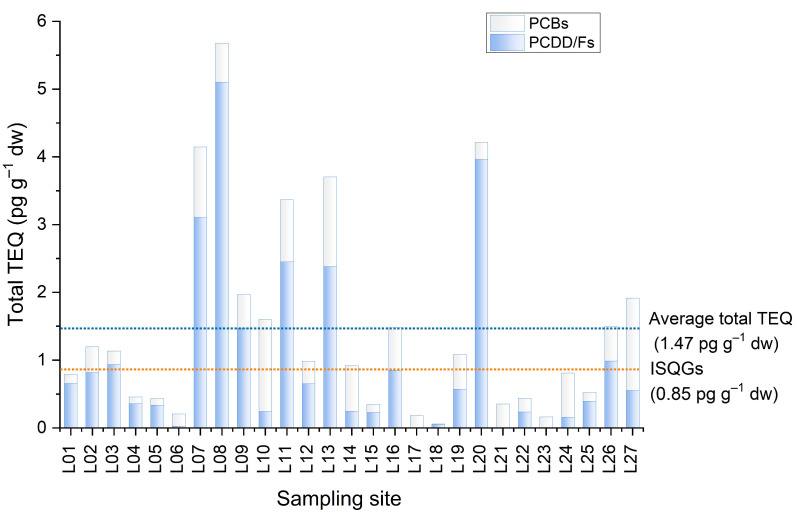
Comparison between the total toxic equivalent quantity (TEQ) of PCDD/Fs and PCBs in the sediments from LSR and the Canadian interim sediment quality guidelines (ISQGs).

**Table 1 ijerph-19-16465-t001:** Concentrations (pg g^−1^ dw) of PCDD/Fs in the surface sediments from LSR.

**Location**	**L01**	**L02**	**L03**	**L04**	**L05**	**L06**	**L07**	**L08**	**L09**	**L10**	**L11**	**L12**	**L13**	**L14**
2,3,7,8-TCDF	nd	0.99	0.71	nd	0.43	nd	4.3	2.6	1.9	nd	1.8	0.97	3.1	nd
1,2,3,7,8-PeCDF	nd	nd	0.54	nd	0.36	nd	1.7	2.1	nd	nd	1.4	nd	1.72	nd
2,3,4,7,8-PeCDF	nd	nd	0.87	0.37	0.23	nd	2.9	5.4	1.3	nd	2.8	0.58	1.4	nd
1,2,3,4,7,8-HxCDF	1.8	1.8	1.3	0.78	0.83	nd	4.1	5.8	2.2	nd	3.8	1.2	3.8	1.0
1,2,3,6,7,8-HxCDF	1.7	1.7	1.4	0.52	0.39	nd	3.6	5.3	1.9	nd	2.7	0.66	2.8	0.51
2,3,4,6,7,8-HxCDF	1.7	1.8	1.3	0.61	0.50	nd	3.6	7.8	2.1	nd	4.4	0.97	3.1	nd
1,2,3,7,8,9-HxCDF	nd	nd	0.35	nd	nd	nd	0.76	1.9	0.37	nd	nd	nd	0.71	nd
1,2,3,4,6,7,8-HpCDF	7.9	9.4	5.8	3.3	2.3	1.3	17.8	31.3	9.0	8.4	14.0	4.7	17.0	4.7
1,2,3,4,7,8,9-HpCDF	0.78	1.1	0.92	0.57	0.36	nd	2.0	3.0	1.6	nd	0.92	0.47	2.3	nd
OCDF	22.0	18.6	9.2	9.1	5.1	2.9	60.5	31.2	31.2	44.6	24.3	11.8	53.4	17.6
ƩPCDFs	35.5	35.3	22.3	15.3	10.5	4.2	101	96.3	51.6	53.0	56.0	21.4	89.3	23.9
2,3,7,8-TCDD	nd	nd	nd	nd	nd	nd	nd	nd	nd	nd	nd	nd	nd	nd
1,2,3,7,8-PeCDD	nd	nd	nd	nd	nd	nd	nd	nd	nd	nd	nd	nd	nd	nd
1,2,3,4,7,8-HxCDD	nd	nd	nd	nd	nd	nd	0.47	1.1	nd	nd	nd	nd	0.33	nd
1,2,3,6,7,8-HxCDD	nd	nd	0.30	nd	nd	nd	1.2	2.7	0.41	nd	nd	nd	0.87	nd
1,2,3,7,8,9-HxCDD	nd	nd	0.24	nd	nd	nd	0.77	1.4	0.35	nd	nd	nd	0.70	nd
1,2,3,4,6,7,8-HpCDD	3.8	6.3	2.8	1.5	1.2	1.0	7.7	20.4	4.0	10.6	12.6	3.4	12.0	3.1
OCDD	22.2	60.8	17.6	7.9	9.8	4.2	45.8	49.0	22.2	132	114	27.1	116	39.4
ƩPCDDs	26.0	67.1	20.9	9.3	11.0	5.2	56.0	74.6	26.9	142	127	30.5	129	42.5
ƩPCDD/Fs	61.5	102	43.2	24.6	21.5	9.4	157	171	78.5	195	183	51.9	219	66.4
ƩPCDDs/ƩPCDFs	0.73	1.90	0.94	0.61	1.05	1.24	0.55	0.77	0.52	2.68	2.27	1.43	1.44	1.78
WHO-TEQ (1998)	0.65	0.80	1.1	0.43	0.39	0.024	3.7	6.2	1.7	0.21	3.0	0.76	2.7	0.24
WHO-TEQ (2005)	0.66	0.82	0.94	0.36	0.34	0.025	3.1	5.1	1.5	0.24	2.4	0.65	2.4	0.25
**Location**	**L15**	**L16**	**L17**	**L18**	**L19**	**L20**	**L21**	**L22**	**L23**	**L24**	**L25**	**L26**	**L27**	
2,3,7,8-TCDF	nd	1.2	nd	nd	0.79	0.54	nd	0.26	nd	nd	nd	1.4	4.5	
1,2,3,7,8-PeCDF	nd	nd	nd	nd	0.45	nd	nd	nd	nd	nd	nd	nd	1.2	
2,3,4,7,8-PeCDF	nd	nd	nd	nd	0.43	nd	nd	0.27	nd	nd	nd	nd	nd	
1,2,3,4,7,8-HxCDF	nd	1.7	nd	nd	0.95	0.95	nd	0.32	nd	0.49	2.6	1.7	nd	
1,2,3,6,7,8-HxCDF	nd	1.6	nd	nd	0.65	1.0	nd	0.36	nd	0.51	nd	2.7	nd	
2,3,4,6,7,8-HxCDF	nd	1.9	nd	nd	0.64	1.3	nd	0.30	nd	nd	nd	2.1	nd	
1,2,3,7,8,9-HxCDF	nd	nd	nd	nd	0.17	nd	nd	nd	nd	nd	nd	nd	nd	
1,2,3,4,6,7,8-HpCDF	8.0	9.2	nd	4.6	3.8	22.7	0.79	1.6	nd	2.2	6.7	11.1	2.6	
1,2,3,4,7,8,9-HpCDF	nd	0.91	nd	nd	0.54	1.0	nd	nd	nd	1.1	nd	1.2	nd	
OCDF	31.8	34.5	nd	8.2	11.9	136	nd	3.8	1.3	7.2	12	35.4	16.1	
ƩPCDFs	39.8	51.1	nd	12.8	20.4	164	0.79	6.9	1.3	11.4	21.2	55.5	24.4	
2,3,7,8-TCDD	nd	nd	nd	nd	nd	nd	nd	nd	nd	nd	nd	nd	nd	
1,2,3,7,8-PeCDD	nd	nd	nd	nd	nd	nd	nd	nd	nd	nd	nd	nd	nd	
1,2,3,4,7,8-HxCDD	nd	nd	nd	nd	nd	1.0	nd	nd	nd	nd	nd	nd	nd	
1,2,3,6,7,8-HxCDD	nd	nd	nd	nd	0.24	4.8	nd	nd	nd	nd	nd	nd	nd	
1,2,3,7,8,9-HxCDD	nd	nd	nd	nd	0.14	1.2	nd	nd	nd	nd	nd	nd	nd	
1,2,3,4,6,7,8-HpCDD	12.7	6.8	nd	nd	2.1	181	nd	1.2	nd	2.3	5.3	5.6	3.5	
OCDD	44.9	70.6	7.1	9.1	13.8	2668	3.0	10.1	2.2	13.2	46.0	41.7	23.5	
ƩPCDDs	57.7	77.5	7.1	9.1	16.2	2856	3.0	11.3	2.2	15.5	51.2	47.2	27.0	
ƩPCDD/Fs	97.5	129	7.1	21.9	36.6	3019	3.8	18.2	3.5	26.9	72.5	103	51.4	
ƩPCDDs/ƩPCDFs	1.45	1.52	–	0.71	0.79	17.41	3.80	1.64	1.69	1.36	2.42	0.85	1.11	
WHO-TEQ (1998)	0.22	0.83	0.0007	0.047	0.66	3.40	0.0081	0.29	0.0004	0.16	0.38	0.97	0.57	
WHO-TEQ (2005)	0.23	0.85	0.0021	0.051	0.57	4.0	0.0088	0.24	0.0011	0.16	0.39	0.99	0.55	

Note: WHO-TEQ, World Health Organization toxic equivalent quantity, $\mathrm{TEQ}=\sum_{i=1}^{n}\left(C_{i}\times {\mathrm{TEF}}_{i}\right)\;$(TEF: toxic equivalency factors; C: the concentration of the individual compound); nd, not detected; –, not available.

**Table 2 ijerph-19-16465-t002:** Concentrations (pg g^−1^ dw) of PCBs in the surface sediments from LSR.

**Location**	**L01**	**L02**	**L03**	**L04**	**L05**	**L06**	**L07**	**L08**	**L09**	**L10**	**L11**	**L12**	**L13**	**L14**
PCB-77	19.5	36.4	12.2	8.1	10.0	4.2	42.3	18.8	24.0	34.3	27.1	12.3	47.0	12.6
PCB-81	1.7	3.3	1.4	0.87	0.91	nd	2.0	1.7	1.7	2.1	2.6	nd	2.5	0.77
PCB-126	1.3	3.7	1.8	0.98	0.94	1.8	9.7	5.2	4.6	12.5	8.4	3.0	12.5	6.2
PCB-169	nd	nd	0.65	nd	nd	nd	1.9	1.5	0.94	3.3	2.3	1.1	1.8	1.6
PCB-105	21.9	48.9	14.1	12.2	14.8	nd	69.0	9.1	30	39.7	38.4	15.0	89.0	22.3
PCB-114	2.6	6.9	2.8	1.4	2.1	0.87	6.9	2.2	4.9	3.2	4.3	2.1	7.1	2.9
PCB-118	38.4	104	38.3	21.2	32.3	3.7	147	24.7	85.7	83.7	93.5	35.9	200	53.4
PCB-123	2.5	3.2	1.2	1.1	1.6	2.9	13.0	6.0	8.3	9.2	12.2	3.8	21.3	6.4
PCB-156	10.4	26.4	7.4	5.4	6.2	1.1	25.6	6.8	12.2	9.8	13.0	5.0	20.4	12.4
PCB-157	2.5	6.0	1.7	0.94	1.2	1.8	13.5	4.0	6.3	20.6	9.8	4.0	13.5	13.9
PCB-167	3.4	5.5	3.0	1.4	0.93	1.8	19.2	5.2	8.8	28.3	12.3	6.8	21.4	19.3
PCB-189	1.7	11.9	1.4	0.68	0.87	0.43	5.5	3.1	2.8	4.5	4.2	33.8	67.9	3.2
Ʃdl-PCBs	106	256	86	54.3	71.9	18.6	356	88.3	190	251	228	122	505	155
PCB-28	217	1211	154	77.1	375	25.1	429	72.7	516	195	402	151	372	33.4
PCB-52	27.9	120	11.3	9.6	27.1	nd	124	12.0	71.0	61.6	68.7	21.3	120	11.3
PCB-101	32.0	78.1	23.7	17.2	17.2	2.8	188	18.0	94.9	106	95.3	42.0	150	66.8
PCB-138	77.3	102	46.5	38.8	40.3	9.3	209	39.4	111	149	140	51.8	213	160
PCB-153	61.5	123	67.7	450	43.7	9.2	258	48.3	127	240	144	65.8	281	190
PCB-180	73.4	111	47.0	26.3	25.6	3.0	134	39.1	76.2	156	99.5	37.6	147	162
ƩIndicator PCBs	489	1745	350	619	529	49.4	1342	230	996	908	950	370	1283	624
PCB-3	22.5	84.6	7.2	2.2	8.8	nd	55.0	28.7	40.6	46.5	41.2	32.2	64.8	107
PCB-15	82.6	138	38.5	27.8	47.5	24.1	113	45.6	76.8	58.4	84.3	42.0	109	37.1
PCB-202	6.2	11.2	3.5	1.2	1.5	2.6	25.6	9.2	16.5	46.0	21.7	8.5	27.4	12.0
PCB-205	1.6	nd	1.2	nd	1.2	nd	2.8	2.4	2.5	5.4	3.5	1.4	3.8	4.0
PCB-208	47.3	40.8	24.8	8.5	7.1	30.8	240	92.4	150	448	188	98.3	293	232
PCB-209	150	216	87.0	126	20.4	193	1835	499	1660	2563	1078	623	1868	1340
ƩPCBs	905	2492	598	839	687	319	3969	996	3132	4326	2595	1297	4154	2511
WHO-TEQ (1998)	0.15	0.41	0.20	0.11	0.10	0.19	1.0	0.55	0.50	1.3	0.90	0.32	1.3	0.66
WHO-TEQ (2005)	0.14	0.38	0.20	0.10	0.10	0.18	1.0	0.57	0.49	1.4	0.92	0.33	1.3	0.67
Total WHO-TEQ (1998)	0.80	1.21	1.31	0.54	0.49	0.21	4.74	6.76	2.22	1.52	3.89	1.09	4.00	0.89
Total WHO-TEQ (2005)	0.79	1.20	1.14	0.46	0.43	0.21	4.14	5.67	1.97	1.60	3.37	0.99	3.70	0.92
**Location**	**L15**	**L16**	**L17**	**L18**	**L19**	**L20**	**L21**	**L22**	**L23**	**L24**	**L25**	**L26**	**L27**	
PCB-77	1.6	27.3	11.0	3.2	20.5	11.0	14.6	4.2	17.1	23.1	14.1	18.9	93.6	
PCB-81	nd	2.1	1.2	nd	3.7	0.73	1.4	nd	1.3	1.6	1.1	2.0	56.4	
PCB-126	1.2	5.7	1.8	nd	4.4	2.3	3.1	1.9	1.5	6.1	1.3	4.5	12.8	
PCB-169	nd	1.8	nd	nd	2.1	0.79	0.86	0.31	0.46	1.2	nd	1.8	1.3	
PCB-105	7.1	46.4	20.6	37.8	76.5	7.4	22.4	1.5	16.0	40.6	16.0	10.6	88.9	
PCB-114	1.3	4.8	3.0	4.1	8.8	1.0	2.9	0.56	2.2	4.0	2.1	2.3	97.5	
PCB-118	36.4	95.7	70.3	164	213	22.1	71.0	4.8	33.8	102	29.5	24.2	226	
PCB-123	3.5	11.0	5.4	10.3	23.1	2.1	9.9	5.1	2.0	14.2	2.3	4.1	25.4	
PCB-156	2.5	13.9	7.3	11.8	16.5	3.9	11.1	1.1	8.7	14.5	8.8	7.7	120	
PCB-157	1.2	8.5	2.9	2.0	7.5	2.7	5.8	2.7	1.7	10.8	1.8	4.7	15.6	
PCB-167	1.4	11.2	4.3	5.8	13.2	3.7	8.3	2.0	2.7	14.2	2.6	5.2	42.5	
PCB-189	0.56	2.5	1.3	1.6	3.1	0.67	2.0	0.53	1.4	3.1	1.4	3.5	34.0	
Ʃdl-PCBs	56.8	231	129	241	392	58.4	153	24.7	88.9	235	81	89.5	814	
PCB-28	36.9	368	209	148	330	127	162	27.9	407	301	377	72.2	872	
PCB-52	3.3	142	33.9	47.6	92.8	15.0	37.0	nd	61.6	63.1	57.1	12.3	83.2	
PCB-101	37.4	124	60.8	192	271	13.7	55.4	7.4	30.6	150	23.0	20.9	260	
PCB-138	43.6	135	91.4	244	293	33.5	90.1	10.0	83.2	167	78.5	45.3	610	
PCB-153	79.3	158	132	394	499	39.9	130	14.0	89.0	222	75.5	49.6	700	
PCB-180	25.4	97.3	70.0	142	184	23.6	71.7	3.8	66.9	124	58.7	36.5	619	
ƩIndicator PCBs	226	1024	597	1168	1670	253	546	63.1	738	1027	670	237	3144	
PCB-3	4.8	33.3	14.7	3.5	51.7	8.4	36.6	nd	nd	nd	nd	12.9	1169	
PCB-15	11.7	81.8	33.4	17.9	64.3	36.4	44.2	24.5	74.9	64.8	72.4	48.2	506	
PCB-202	2.8	19.9	25.3	1.5	22.4	6.7	11.6	2.7	2.4	26.9	2.6	9.4	9.5	
PCB-205	0.84	2.7	2.0	1.3	6.0	1.4	1.6	0.48	0.97	2.9	1.2	2.7	5.5	
PCB-208	21.4	177	49.2	9.5	168	73.7	104	33.3	10.3	254	9.9	95.8	18.3	
PCB-209	115	1269	292	33.4	953	627	729	211	273	1325	246	522	282	
ƩPCBs	439	2839	1143	1475	3328	1065	1627	360	1189	2936	1083	1017	5949	
WHO-TEQ (1998)	0.12	0.62	0.19	0.031	0.51	0.24	0.34	0.20	0.16	0.65	0.14	0.48	1.5	
WHO-TEQ (2005)	0.12	0.63	0.18	0.0074	0.51	0.25	0.35	0.20	0.16	0.65	0.13	0.51	1.4	
Total WHO-TEQ (1998)	0.34	1.45	0.20	0.078	1.17	3.64	0.35	0.49	0.16	0.81	0.52	1.45	2.03	
Total WHO-TEQ (2005)	0.36	1.48	0.18	0.058	1.09	4.21	0.35	0.44	0.16	0.81	0.53	1.49	1.92	

Note: dl-PCBs, dioxin-like PCBs; total WHO-TEQ, sum of PCDD/Fs and PCBs; nd, not detected.

## Data Availability

The data presented in this study are available in the article and the Appendix A.

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
