# Peer review of "Occurrence and Source Identification of Polychlorinated Dibenzo-p-dioxins and Dibenzofurans and Polychlorinated Biphenyls in Surface Sediments from Liangshui River in Beijing, China"

_ijerph, 2022, doi:10.3390/ijerph192416465_

Round 1

Reviewer 1 Report

The manuscript reports the results of a survey on presence, typology, origin and threat of PCDD/Fs and PCBs contamination in the sediments of the Liangshui River, whose waters are also used for irrigation, hence the risk for the environment and human health.

the text is clear, relevant for the field and presented in a detailed and well-structured style.

the manuscript is scientifically rigorous, and the experimental design is appropriate to test the hypothesis.

Materials and methods are accurately reported, so the tests are reproducible for interested researchers

Figures and tables are appropriate; tables could be more reader-friendly if included in the text, as they are the core of the trial and the bases of discussion; on the contrary, the shift between main text and supplements could be distractive for the reader

Conclusions are coherent with presented results.

Cited references are current, pertinent, and not excessively self-citing.

Author Response

Response to Reviewer 1 Comments

Comments and Suggestions for Authors

The manuscript reports the results of a survey on presence, typology, origin and threat of PCDD/Fs and PCBs contamination in the sediments of the Liangshui River, whose waters are also used for irrigation, hence the risk for the environment and human health.

The text is clear, relevant for the field and presented in a detailed and well-structured style.

The manuscript is scientifically rigorous, and the experimental design is appropriate to test the hypothesis.

Materials and methods are accurately reported, so the tests are reproducible for interested researchers.

Conclusions are coherent with presented results.

Cited references are current, pertinent, and not excessively self-citing.

Point 1: Figures and tables are appropriate; tables could be more reader-friendly if included in the text, as they are the core of the trial and the bases of discussion; on the contrary, the shift between main text and supplements could be distractive for the reader.

Response: We thank the reviewer for the comments and kind suggestion. Table S1 and Table S2 have been placed directly in the main paper.

Reviewer 2 Report

The followings are necessary to improve the quality of the paper.

In the abstract and discussion section: is WHO-TEQ is for human exposure pathways? If yes, the authors should explain how river sediments are likely to impact human. The ISQGs seems to be a better benchmark.

Line 50: Authors should provide the reference for the hydrophobic properties.

Line 54 - 56: is this statement describing the river? if yes, what do you mean by "population of 4.5 million"? (in the river?)

Materials and methods: Authors did not provide information on the depth of the river at L01 - L27. THIS IS VERY IMPORTANT. Authors is encouraged to provide the concentrations of PCDD/Fs and PCBs as a function of depth. Sallow regions of the river will be exposed to more sunlight UV (than deep regions) and the contaminants will degrade at a relatively faster rate due to photodegradation (REF1 & 2) and singlet oxygen (REF3). See REF1, REF2, REF 3 as a guide.

**REF1: Adriana R. Nicolaescu, Photodegradation of PCBS in Natural Waters, Western Michigan University Thesis,

**REF2: Lin, Y., Gupta, G., & Baker, J. (1995). Photodegradation of polychlorinated biphenyl congeners using simulated sunlight and diethylamine. Chemosphere, 31(5), 3323-3344.

**REF3: Zeinali, Nassim, et al. "Destruction of dioxin and furan pollutants via electrophilic attack of singlet oxygen." Ecotoxicology and Environmental Safety 184 (2019): 109605.

Line 120-125: what is the effect of granulated copper on catalytic degradation of PCDD/F and PCB ? Was the copper present during the extraction at 150 C? This temperature is high to decompose the  contaminants in the presence of granulated copper.

Result and discussion: the results are well discussed. However, authors should consider if the effect of depth (UV attenuation) is significant. Also, authors should clarify how the Total TeQ in Figure 5 was calculated. If possible, Figure S2 should be placed (and discussed) directly in the main paper.

Author Response

Response to Reviewer 2 Comments

Comments and Suggestions for Authors

The followings are necessary to improve the quality of the paper.

Point 1. In the abstract and discussion section: is WHO-TEQ is for human exposure pathways? If yes, the authors should explain how river sediments are likely to impact human. The ISQGs seems to be a better benchmark.

Reply: Thanks for the comments. In this study, the concentrations of PCDD/Fs and PCBs were expressed as both pg g–1 and pg TEQ g–1. The WHO-TEQ concentration was used to evaluate the health risks of exposures to contaminations in various matrices according to the interim sediment Quality Guidelines (ISQGs) derived by Canadian Council of Ministers of the Environment (CCME).

According to the suggestion, we have revised the abstract and discussion section:

(1) Abstract part, paragraph 1: We have changed “Due to their hydrophobic properties, these compounds are strongly adsorbed on particulate matters in water body, and then deposited to sediment” to “Due to their hydrophobic properties [9], these compounds are strongly adsorbed on particulate matters in water body, and mostly deposited into sediments. Aquatic organisms are exposed to these compounds through direct contact and ingestion, which may pose a potential threat to aquatic organisms and even human health through aquatic food chains”.

(2) Abstract part, paragraph 3: We have changed “LSR basin is the main agricultural area and the largest wastewater irrigation area in Beijing. Pollutants would inevitably transport from aquatic environment to land [14–16], and subsequently threaten the local vegetation, livestock and residents” to “Owing to water shortage in the megacity Beijing, wastewater is still one of the important non-conventional sources for many rivers such as LSR, and therefore is used widely for agricultural purposes by farmers along many rivers. LSR basin, the main agricultural area in Beijing, is such a wastewater irrigation area. Pollutants in aquatic environment would be inevitably transferred to land [10–12] through irrigation, and subsequently to crops and humans”.

(3) We have also reworded the second paragraph in section 3.6 (discussion part) to explain the health exposure risk, please find the details in the manuscript.

Point 2. Line 50: Authors should provide the reference for the hydrophobic properties.

Reply: According to the reviewer comment, we have provided the reference for the hydrophobic properties in the abstract, paragraph 1, “Due to their hydrophobic properties [9], these compounds are strongly adsorbed on particulate matters in water body”.

Point 3. Line 54 - 56: is this statement describing the river? if yes, what do you mean by "population of 4.5 million"? (in the river?)

Reply: The details of the river were presented in the abstract, paragraph 2. "population of 4.5 million" means the population living in Liangshui River basin. According to the comments from the reviewer, we have replaced the sentence “It stretches 68 kilometers with a basin extending over 600 square kilometers and a population of 4.5 million inhabitants” with “It stretches 68 kilometers with a basin extending over 600 square kilometers”.

Point 4. Materials and methods: Authors did not provide information on the depth of the river at L01 - L27. THIS IS VERY IMPORTANT. Authors is encouraged to provide the concentrations of PCDD/Fs and PCBs as a function of depth. Sallow regions of the river will be exposed to more sunlight UV (than deep regions) and the contaminants will degrade at a relatively faster rate due to photodegradation (REF1 & 2) and singlet oxygen (REF3). See REF1, REF2, REF 3 as a guide.

**REF1: Adriana R. Nicolaescu, Photodegradation of PCBS in Natural Waters, Western Michigan University Thesis,

**REF2: Lin, Y., Gupta, G., & Baker, J. (1995). Photodegradation of polychlorinated biphenyl congeners using simulated sunlight and diethylamine. Chemosphere, 31(5), 3323-3344.

**REF3: Zeinali, Nassim, et al. "Destruction of dioxin and furan pollutants via electrophilic attack of singlet oxygen." Ecotoxicology and Environmental Safety 184 (2019): 109605.

Reply: Thanks for the comments. According to the suggestion, we have added the information in part 2.1, “The sampling campaign was carried out in the flood season of 2013, and the depth of LSR was up to 3 m”.

We appreciated the authors’ consideration about possible degradation of POPs in the surface sediment, although these pollutants in sediments are generally chemically stable and the laboratory incubation studies suggest that their photolysis, hydrolysis, or microbial degradation in aquatic sediments are minor [1]. However, this study aims to evaluate occurrence and spatial distribution of these compounds, and further identify the possible sources and access the potential exposure risk. The sampling methods for PCDD/Fs and PCBs in sediments in the present study strictly followed the national standards and international guidelines, such as HJ 77.4-2008 [2],US EPA 1613B [3] and US EPA 1668C [4]. The methods have been widely employed in many authoritative studies [5-9] (Table S1 in the supplementary materials provided more references). Therefore, in term of the objective of this study, we do not think it is obligatory to evaluate the relationship between the concentrations of POPs and sampling depth, since the spatial distribution of POPs in the sediments may vary widely along with the distance from sources.

[1] CCME 2001. Canadian sediment quality guidelines for the protection of aquatic life, polychlorinated dibenzo-p-dioxins and polychlorinated dibenzofurans (PCDD/Fs), Canadian Council of Ministers of the Environment.

[2] HJ 77.4-2008. Soil and sediment  Determination of polychlorinated dibenzo-p-dioxins (PCDDs) and polychlorinated dibenzofurans (PCDFs)  Isotope dilution HRGC-HRMS.

[3] US EPA, 1997. Method 1613B. Tetra- through octa-chlorinated dioxins and furans by isotope dilution HRGC/HRMS.

[4] US EPA, 2010. Method 1668C. Chlorinated biphenyl congeners in water, soil, sediment, biosolids and tissue by HRGC/HRMS.

[5] Ishaq R.; Persson N.J.; Zebühr Y.; Broman D. PCNs, PCDD/Fs, and non-orthoPCBs, in water and bottom sediments from the industrialized Norwegian Grenlandsfjords. Environ. Sci. Technol. 2009, 43, 3442–3447.

[6] Cole J.G.; Mackay D.; Jones K.C.; Alcock R.E. Interpreting, correlating, and predicting the multimedia concentrations of PCDD/Fs in the United Kingdom. Environ. Sci. Technol. 1999, 33, 399–405.

[7] Green N.J.L.; Jones J.L.; Jones K.C. PCDD/F deposition time trend to Esthwaite Water, U.K., and its relevance to sources. Environ. Sci. Technol. 2001, 35, 2882–2888.

[8] Howell N.L.; Suarez M.P.; Rifai H.S.; Koenig L. Concentrations of polychlorinated biphenyls (PCBs) in water, sediment, and aquatic biota in the Houston Ship Channel, Texas. Chemosphere 2008, 70, 593–606.

[9] Li A.; Guo J.; Li Z.; Lin T.; Zhou S.; He H.; Ranansinghe P.; Sturchio N.C.; Rockne K.J.; Giesy J.P. Legacy polychlorinated organic pollutants in the sediment of the Great Lakes. J. Great Lakes Res. 2018, 44, 682–692.

Point 5. Line 120-125: what is the effect of granulated copper on catalytic degradation of PCDD/F and PCB? Was the copper present during the extraction at 150 C? This temperature is high to decompose the contaminants in the presence of granulated copper.

Reply: The granulated copper was used to remove the elemental sulfur in the extracted solution after ASE extraction. It wasn’t added before the extraction. According to the reviewer comments, to avoid misunderstanding, we have reworded the sentence as follows: “After extraction, granulated activated copper was added in the receiving flasks to remove elemental sulfur”.

Point 6. Result and discussion: the results are well discussed. However, authors should consider if the effect of depth (UV attenuation) is significant. Also, authors should clarify how the Total TeQ in Figure 5 was calculated. If possible, Figure S2 should be placed (and discussed) directly in the main paper.

Reply: We thank the reviewer for the kind suggestion. We have added the formula for calculating TEQ in the note of Table 1, “ (TEF: toxic equivalency factors; C: the concentration of the individual compound)”; In part 3.6, we have changed “Figure 5 showed the ISQGs and the sum TEQ levels of PCDD/Fs and PCBs in our study” to “Figure 6 showed the ISQGs and the total TEQ (the sum TEQ levels of PCDD/Fs and PCBs)”; Figure S2 have been placed in the main paper; Concerning the effect of depth, we have demonstrated in the response to Point 4.

Reviewer 3 Report

Dear authors,

Congratulations for your work.

Please find below some suggestions for paper improving:

1. Not all the not original information, data, information, especially in introduction, but not only, have the needed citations and references.

2. Please highlight the importance of your study.

3. The discussion and conclusions can be improved to highlight the need of this study in a more holistic approach of these substances effects over the environment and public health.

4. Highlight more the need and importance of the original obtained data for stakeholders, managers and state institutions.

Author Response

Response to Reviewer 3 Comments

Comments and Suggestions for Authors

Dear authors,

Congratulations for your work.

Please find below some suggestions for paper improving:

Point 1. Not all the not original information, data, information, especially in introduction, but not only, have the needed citations and references.

Reply: We thank the reviewer for the kind suggestions. We have deleted 13 unnecessary citations and references in the introduction, part 3.1 and references.

Point 2. Please highlight the importance of your study.

Reply: We have improved the introduction to highlight the importance of this study according to the reviewer suggestions, please find the details in the introduction, paragraph 3.

Point 3. The discussion and conclusions can be improved to highlight the need of this study in a more holistic approach of these substance’s effects over the environment and public health.

Reply: According the suggestions from the reviewer, we have improved section 3.6 (paragraph 1 and paragraph 2) and the conclusion (paragraph 2) to highlight the need of this study, please find the details in the manuscript.

Point 4. Highlight more the need and importance of the original obtained data for stakeholders, managers and state institutions.

Reply: We thank the reviewer for the kind suggestions. We have improved the conclusion and highlighted the need and importance of the data for stakeholders, managers and state institutions, please find the details in the conclusion, paragraph 2.

Reviewer 4 Report

Honghua Li et al. reported the manuscript with titled “Occurrence and source identification of PCDD/Fs and PCBs in surface sediments from Liangshui River in Beijing, China”. The reported data is not bad and also publishable in present journal. But before acceptance the manuscript must revise carefully with some minor changes.

Comments and Suggestion. 

1.      The title must be revised without abbreviations. Its will be very helpful for the reader because most of the reader don’t know about it in initial sight.

2.      Don’t use abbreviation in abstract part. Please make this habit that don’t use abbreviation in your next manuscripts.

3.      Improve the English throughout the manuscript.

4.      Please highlight the effect of such hydrophobic pollutants on aquatic organisms in introduction part.

5.      July 2013, so old data? what does it mean?

6.      Page 3 line 102, What is mean by transported? Rephrase this sentence.

7.      Mention the purity of the chemicals in chemical part.

8.      Check the manuscript thoroughly for spelling mistakes.

Author Response

Response to Reviewer 4 Comments

Comments and Suggestions for Authors

Honghua Li et al. reported the manuscript with titled “Occurrence and source identification of PCDD/Fs and PCBs in surface sediments from Liangshui River in Beijing, China”. The reported data is not bad and also publishable in present journal. But before acceptance the manuscript must revise carefully with some minor changes.

Comments and Suggestion.

Point 1. The title must be revised without abbreviations. Its will be very helpful for the reader because most of the reader don’t know about it in initial sight.

Reply: According to the comments from the reviewer, we have used “polychlorinated dibenzo-p-dioxins and dibenzofurans and polychlorinated biphenyls” to replace “PCDD/Fs and PCBs” in the title.

Point 2. Don’t use abbreviation in abstract part. Please make this habit that don’t use abbreviation in your next manuscripts.

Reply: We thank the reviewer for the kind suggestion. The abbreviations in the abstract have been deleted, and we have provided them in other parts of the main text.

Point 3. Improve the English throughout the manuscript.

Reply: We have improved the English throughout the manuscript according to the suggestions.

Point 4. Please highlight the effect of such hydrophobic pollutants on aquatic organisms in introduction part.

Reply: According to the reviewer suggestions,we have changed “Due to their hydrophobic properties [33], these compounds are strongly adsorbed on particulate matters in water body, and then deposited to sediment” to “Due to their hydrophobic properties [9], these compounds are strongly adsorbed on particulate matters in water body, and mostly deposited into sediments. Aquatic organisms are exposed to these compounds through direct contact and ingestion, which may pose a potential threat to aquatic organisms and even human health through aquatic food chains”

Point 5. July 2013, so old data? what does it mean?

Reply: Sample collection was conducted in 2013, and sample preparation and instrumental analysis were finished in the next year. However, due to the delay for manuscript preparation, the submission was put on a schedule just recently.

Point 6. Page 3 line 102, What is mean by transported? Rephrase this sentence.

Reply: We have rephrased the sentence (part 2.1, paragraph 1), “All the samples were sealed in clean polyethylene bags, carried to the laboratory immediately, and then stored at –18 °C”.

Point 7. Mention the purity of the chemicals in chemical part.

Reply: We have added the purity of the chemicals in section 2.2, paragraph 1, “13C12-labeled standard solutions of PCDD/Fs (purity > 99.9%)”, “PCBs (purity > 99.9%)”.

Point 8. Check the manuscript thoroughly for spelling mistakes.

Reply: Thanks for the fine suggestions. We have checked the manuscript and corrected the spelling mistakes.

Round 2

Reviewer 3 Report

Dear authors,

Congratulation for the work.

Still there are phrases, which not contain authors original results in the text without any citation and references. May be you can add them.

All the best

Author Response

Response to Reviewer 3 Comments

Point 1: Still there are phrases, which not contain authors original results in the text without any citation and references. May be you can add them.

Response 1: We really appreciate the reviewer for the kind suggestion. In Figure 4 and Figure 5, the original data of the typical sources were from some references provided in the supplementary materials (Table S2 and Table S3). According to the reviewers’ suggestion, we have placed these citations in the titles of Figure 4 and Figure 5, and added the corresponding references in the references part of the main paper. The supplementary materials have been revised accordingly.
